# Specific Receptors for the Chemokines CXCR2 and CXCR4 in Pancreatic Cancer

**DOI:** 10.3390/ijms21176193

**Published:** 2020-08-27

**Authors:** Ala Litman-Zawadzka, Marta Łukaszewicz-Zając, Mariusz Gryko, Agnieszka Kulczyńska-Przybik, Bogusław Kędra, Barbara Mroczko

**Affiliations:** 1Department of Neurodegeneration Diagnostics, Medical University of Bialystok, 15-269 Bialystok, Poland; agnieszka.kulczynska-przybik@umb.edu.pl (A.K.-P.); mroczko@umb.edu.pl (B.M.); 2Department of Biochemical Diagnostics, Medical University of Bialystok, 15-269 Bialystok, Poland; marta.lukaszewicz-zajac@umb.edu.pl; 3Second Department of General Surgery, Medical University of Bialystok, 15-276 Bialystok, Poland; mariusz_gryko@vp.pl (M.G.); chgastro@umb.edu.pl (B.K.)

**Keywords:** pancreatic cancer, receptors for chemokines, biomarker

## Abstract

Background: The mortality rate of pancreatic cancer (PC) is equal to its incidence and the majority of PC patients die within a few months of diagnosis. Therefore, a search for new biomarkers useful in the diagnosis and prognosis of PC is ongoing. Objectives: The aim of our study was to compare the utility of CXCR2 and CXCR4 in the diagnosis and prediction of PC with classical tumor marker (carcinoembryonic antigen, CEA) and marker of inflammation–C-reactive protein (CRP). Patients and Methods: The study comprised 64 subjects — 32 PC patients and 32 healthy volunteers. Serum concentrations of tested proteins were analysed using immunological methods. Results: Serum CXCR2 and CXCR4 concentrations, similarly to those of CEA and CRP, were significantly elevated in PC patients compared to healthy controls. Moreover, concentrations of CXCR4 were significantly correlated with CXCR2 and CRP levels, while CRP concentrations were correlated with CXCR2 and CEA levels. The diagnostic sensitivity and the predictive value for negative (PV_−ve_) results for CXCR4 were similar to those of CEA and higher than those of CXCR2 and CRP, while the area under the ROC curve (AUC) for CXCR4 was the highest among all tested proteins (CXCR2, CEA, CRP). Moreover, serum CXCR2 was found to be a significant predictor of PC risk. Conclusions: CXCR4 is a better candidate for a tumor marker than CXCR2 in the diagnosis of PC, while serum CXCR2 is a significant predictor of PC risk.

## 1. Introduction

Chemokines are a family of chemotactic cytokines that bind to their cognate G-protein- coupled receptors in order to cause a cellular response such as adhesion, chemotaxis and migration. These proteins are produced by various tissue cells and leukocytes and therefore are able to regulate leukocyte migration in inflammation and immunity processes [1,2,3,4]. A number of previous studies have investigated if C-X-C motif chemokine receptor (CXCR) 2 and C-X-C motif chemokine receptor (CXCR) 4 play an important role in tumor proliferation, invasion, angiogenesis, metastasis and migration in numerous types of malignancies, including pancreatic cancer (PC) [5,6,7,8,9,10,11,12,13]. 

The N-terminal domain of specific receptors is extracellular and is thus involved in chemokine binding. CXCR2 is widely expressed by neutrophils and endothelial cells. All ELR+ CXC chemokine ligands (CXCL1, CXCL2, CXCL3, CXCL5, CXCL6, CXCL7, CXCL8) bind to CXCR2. However, the most potent ligand of this specific receptor is C-X-C motif chemokine 8 (CXCL8) and cleavage products of this chemokine [9,10,11,12,13,14,15,16,17,18,19,20]. It has been shown that CXCR2 is a pro-tumorigenic chemokine receptor that might induce inflammation in the tumor microenvironment [9,10]. Furthermore, interaction between CXCR2 and tumor microenvironment has been demonstrated to be of critical importance for tumor progression [21]. 

The most extensively studied receptor is CXCR4, which is one of C-X-C motif chemokine 12 (CXCL12) specific receptors [7]. Some clinical investigations have proved that CXCR4 overexpression in cancer cells contributes to tumor growth, invasion, angiogenesis and metastasis [7]. The majority of studies concerning this issue have been performed on PC tissue samples to assess the expression of these proteins in PC cells. The expression of CXCR2 was higher in PC tissue than in non-PC samples and therefore the authors concluded that PC cells have an enhanced capability for producing chemokines in comparison with inflammatory cells [22]. Moreover, the expression rate of CXCR4 in PC cells was 100%, whereas it was totally negative in non-PC controls. The authors revealed that the overexpression of CXCR4 significantly correlated with a more advanced tumor stage and PC progression [23,24,25]. 

Pancreatic cancer (PC) is the fourth most common cause of cancer-related mortality [26,27]. PC is commonly diagnosed at an advanced stage even if the diagnosis is established immediately after symptom onset [26,27]. Despite recent advances in surgical techniques, radiation therapy and chemotherapy, a 5-year survival rate stands at 8% [28]. Computed tomography (CT) is a first-line imaging modality for suspected PC, while magnetic resonance cholangiopancreatography (MRCP) is a second-line modality for suspected PC. In clinical practice, CT and magnetic resonance (MR) are highly effective at detecting this malignancy. However, due to their limitations such as the high cost or invasive nature, other, inexpensive and easy to perform, methods of PC diagnosis are sorely needed. In clinical practice, carbohydrate antigen 19-9 (CA 19-9) and carcinoembryonic antigen (CEA) have been established as reliable serum markers for PC. However, the clinical utility of classical tumor markers is unsatisfactory due to their low diagnostic sensitivity and specificity, particularly in the early stages of PC [29]. 

The presence of soluble chemokine receptors in sera is a general phenomenon that occurs under normal physiological conditions. It is now well established that soluble chemokine receptors may play an important role in human diseases, and their biological effects are already used in therapies [30,31]. The majority of the soluble chemokine receptors are released from the cell surface by proteolytic cleavage. It is also suggested that two in vivo mechanisms for the generation of soluble receptors are known: shedding of membrane-bound receptors and gene expression [32]. Some clinical investigation have revealed that loss of membrane expression of CXCR2 from neutrophils leads to production of soluble acidic glycopeptide CXCR2 [33]. It is suggested that shedding of membrane-bound receptors, including CXCR2 occurs via proteolytical cleavage to soluble forms by cell-associated metalloproteinases. In addition, the study of Malvoisin et al. [34] has described the existence of a soluble form of CXCR4 in human sera using isoelectric focusing (IEF) and Western blotting. Authors performed IEF on agarose gel to demonstrate the existence of a soluble form of CXCR4 in human sera and confirmed that serum CXCR4 level was significantly elevated in cancer patients such as colorectal cancer patients compared to controls. Moreover, some anti-CXCR4 antibodies recognized by western blotting serum CXCR4 in several samples, including those of control subjects. The authors concluded that CXCR4 might be a potential biomarker, particularly of inflammatory diseases such cancer. Therefore, based on the results of other authors as well as the findings of our previous studies performed on esophageal cancer patients [35,36], we decided to assess serum concentrations of selected receptors for chemokines (CXCR2 and CXCR4) in order to determine whether these proteins might be used as potential diagnostic and prognostic biomarkers in PC. To the best of our knowledge, the present study is the first to compare the clinical relevance of serum concentrations of CXCR2 and CXCR4 in relation to the well-established tumor marker (carcinoembryonic antigen–CEA) in PC patients and the marker of inflammation–C-reactive protein (CRP). The present paper is a continuation of our previous investigations, in which we assessed whether serum levels of selected chemokines and their specific receptors might be used as potential tumor biomarkers for gastrointestinal malignancies [35,36,37,38].

## 2. Patients and Methods 

Our inclusion criteria were as follows: patients with pancreatic ductal adenocarcinoma (20 men and 12 women, aged 44–82 years) before treatment diagnosed in the Second Department of General Surgery, Medical University of Bialystok, Poland. Microscopic examination of material obtained during biopsy and/or surgery was used for the clinical diagnosis of PC. All tumors were staged in accordance with the tumor-nodulus-metastases (TNM) classification, proposed by the 5th International Union Against Cancer (UICC) [39]. The control group included 32 healthy volunteers (17 men and 15 women, aged 27–67 years), who were recruited from hospital volunteer organizations. Characteristics of PC patients are presented in Table 1.

All PC patients participating in the study were divided into groups according to the following criteria: tumor stage (I + II, III, and IV), depth of tumor invasion (T1 + T2 + T3, and T4), presence of lymph node metastasis (N0 and N1) and distant metastasis (M0 and M1). The study was approved by the Local Ethics Committee (R-I-002/65/2017) and all the participants signed informed consent forms.

Blood samples from all the patients were obtained prior to treatment (Sarstedt, Nümbrecht, Germany) and stored at −80 °C until analysis. Serum CXCR2 and CXCR4 concentrations were measured using enzyme-linked immunosorbent assay kits (ELISA) (EIAab, Wuhan, China) according to the manufacturer’s instructions. Levels of the well-established tumor marker (CEA) were measured in the sera of patients by the chemiluminescent microparticle immunoassay (CMIA) method (Abbott Laboratories, Abbott Park, IL, USA) using ARCHITECT 8200 ci. Serum CRP levels were examined by the immunoturbidimetric method (Abbott) using ARCHITECT 8200 ci. In order to select optimal predicted probability cut-off values, the Youden’s index was used, because this index reflects the intension to maximize the correct classification rate. The reference cut-off values were 0.72 ng/mL for CXCR2; 1.56 ng/mL for CXCR4; 3.20 mg/L for CRP and 1.24 ng/mL for CEA.

### Statistical Analysis

Serum CXCR2, CXCR4, CEA and CRP levels did not follow a normal distribution in the preliminary statistical analysis (χ^2^-test) and therefore nonparametric statistical analyses were performed. The Mann-Whitney test was used to compare two groups, while the Kruskal-Wallis test was employed in the analysis of three or more groups. In addition, the post hoc Dwass-Steele-Critchlow-Fligner test was performed if significant differences were calculated [40]. Moreover, for correlation analyses the Spearman rank correlation test was utilised. Furthermore, diagnostic sensitivity and specificity, accuracy, predictive value for positive (PV_+ve_) and negative (PV_−ve_) results as well as AUC were also calculated to assess the diagnostic utility of the tested proteins. The differences were considered as statistically significant when *P* < 0.05. IBM SPSS Statistics 20.0 was employed for statistical analysis, while Microsoft Office Excel (company, city, state abbrev if USA, country) was used to calculate diagnostic parameters. Logistic regression was performed to evaluate the strength of association between various risk factors and PC. In the first step, univariate logistic regression models were performed to evaluate the relationship of each variable with PC risk. Then, variables with *p* < 0.05 were introduced into the multivariate model. In the last step of analysis, the significant variables were removed in a stepwise manner from the model based on the Wald statistic.

## 3. Results 

Serum concentrations of both specific receptors for chemokines (CXCR4, *p* < 0.001 and CXCR2, *p* = 0.01) as well as the classical tumor marker (CEA, *p* < 0.001) and the marker of inflammation (CRP, *p* < 0.001) were statistically significantly higher in PC patients when compared to healthy volunteers (Table 2).

When we considered the association between serum concentrations of the analyzed proteins and PC tumor stage according to TNM classification, we found that the serum levels of CXCR4, CRP and CEA were higher in advanced stage of disease in comparison to early PC. However, there were no differences of CXCR2 and CXCR4 levels between cancer patients and stage of the disease (Table 3). 

If we analyzed associations between serum concentrations of the analyzed proteins and the depth of tumor invasion (T-factor) in PC, we found that serum levels of CXCR4, CXCR2 and CEA were higher in the T4 subgroup in comparison to T1 + T2 + T3 subgroup, however these differences were not statistically significant (Table 4). In addition, we established that serum CXCR4 levels were higher in patients with nodal involvement (N1 subgroup) and distant metastasis (M1 subgroup) in comparison to patients without nodal involvement (N0 subjects) and the presence of distant metastasis (M0 subjects), similarly to serum CRP and CEA levels, but statistically significant differences were not found (Table 4).

The Spearman rank correlation test was performed to assess correlation analyses (Table 5). The concentrations of CXCR4 significantly correlated with CXCR2 (*p* < 0.001) and CRP (*p* = 0.01), while CRP concentrations correlated with CXCR2 (*p* = 0.03) and CEA (*p* < 0.001) concentrations (Table 5).

The percentage of elevated results (diagnostic sensitivity) for CXCR4 (91%) was higher than for CXCR2 (75%) and CRP (69%), but marginally lower than for CEA (94%). In addition, the combined analysis of CXCR4 and CEA or CRP improved the diagnostic sensitivity up to 100% (Figure 1). The diagnostic specificity for CXCR4 levels (69%) was higher than that for CXCR2 (66%) and the classical tumor marker (CEA–47%), but lower in comparison to CRP levels (94%), similarly to the predictive value for positive (PV_+ve_) results. However, the predictive value for negative (PV_−ve_) results for CXCR4 (88%) was the same as for CEA and higher than that of CXCR2 (72%) and CRP (75%). The diagnostic accuracy of CXCR4 (80%) was similar to that of CRP (81%), but far higher than that of CXCR2 and CEA (70%). The highest accuracy was observed for the combined use of CXCR4 and CRP (83%) (data not shown). 

The AUC for CXCR4 (0.8496, *p* < 0.001) was the highest among all the proteins tested (CXCR2–0.7373, *p* = 0.001), CEA (0.7539, *p* < 0.001) and CRP (0.8013, *p* < 0.001) (Figure 2). The cut-off values of the analysed proteins were established using the Youden Index and were as follows: 0.72 ng/mL for CXCR2; 1.56 ng/mL for CXCR4; 3.20 mg/L for CRP and 1.24 ng/mL for CEA.

In the first step of analysis, the association between PC risk and several risk factors was analysed in univariate analysis in order to identify risk factors that qualify for the multivariate model. The results were presented as *p* value as well as odd ratios (OR). The serum levels of CXCR4 (*p* = 0.01 OR = 1.228), CXCR2 (*p* = 0.01, OR = 5.816), the well-established tumor marker–CEA (*p* = 0.01, OR = 1.684) and CRP (*p* = 0.01, OR = 1.493) as well as patients age (*p* < 0.001, OR = 1.137) were associated with a significantly increased prediction of PC. In the next step of our analysis, the variables which were found to be statistically significant in the univariate logistic regression analysis were entered into the multivariate model. Finally, in the last step of analysis, significant variables were removed from the model in a stepwise manner. Therefore, in the final model only serum CXCR2 (*p* = 0.05, OR = 4.618) and CEA (*p* = 0.01, OR = 1.807) as well as age (*p* = 0.01, OR = 1.135) were significant predictors of PC risk.

## 4. Discussion

PC is characterised by abnormal expression of various growth factors, enhanced angiogenesis and resistance to apoptosis, all of which impact on the aggressiveness of PC [41,42]. Chemokines and their specific receptors play a crucial role in tumor development, including PC [43]. PC is a highly invasive disease as 80% of patients present with tumors extending locally beyond the pancreas as well as metastases. Furthermore, the aggressive nature of the tumor and a lack of sensitivity to most treatment modalities are the primary cause of PC mortality, with a rate equal to its incidence [43]. Therefore, an intensive search for new biomarkers useful in the diagnosis and prognosis of PC is ongoing. The role of specific receptors for chemokines in the development of various malignancies, including PC, has been suggested by some authors, but the studies were commonly performed on PC tissue, using mostly immunohistochemical techniques [5,8,17,24,25]. To our knowledge, there is a dearth of research regarding serum CXCR2 and CXCR4 concentrations in PC patients in relation to the clinicopathological characteristics of the tumor as well as the diagnostic and prognostic potential of these protein in comparison to the classical tumor biomarker (CEA) and the marker of inflammation (CRP). The present paper is a continuation of our previous studies concerning the search for novel biomarkers of PC, such as selected cytokines [44] including chemokines [38] as well as several metalloproteinases (MMP-9, MMP-2) and their tissue inhibitors [45,46]. 

In the present paper we demonstrated that serum CXCR4 and CXCR2 were significantly elevated in PC patients compared to healthy controls, similarly to the classical tumor marker and CRP. Based on our results, we may suggest that PC cells are able to produce these proteins. Other researchers have also revealed that the expression of CXCR2 and CXCR4 in PC tissue is higher in comparison to non-PC controls. The authors have proved that PC cells possess a higher capability for producing chemokines than inflammatory cells [24,43,47,48].

Our study found significant correlations between serum concentrations of the analysed proteins and the clinicopathological characteristics of PC. However, significant differences were indicated only between serum levels of the analysed receptors (CXCR4 and CXCR2) in all the studied subgroups of PC patients (tumor stage, T, N, M factor) and control groups. Contradictory results have been presented by other authors who revealed that CXCR4 expression significantly correlated with advanced stages of the disease and tumor progression [24,25]. In addition, a study by Marchesi et al. indicated that the CXCR4 receptor is frequently expressed in metastatic PC cells [49]. Our previous study revealed that CXCL8 concentrations were significantly higher in patients with the presence of lymph node metastasis in comparison to patients without nodal involvement [38].

In our present study we performed the Spearman rank correlation test. The concentrations of CXCR4 significantly correlated with CXCR2 and CRP levels, while CRP concentrations correlated with CXCR2 and CEA levels. We have previously demonstrated that serum CXCL8 levels significantly correlated with CRP levels and the presence of lymph node metastasis in PC patients [38].

The present study evaluated the diagnostic significance of selected receptors for chemokines (CXCR4 and CXCR2) in comparison to the well-established tumor marker–CEA. The diagnostic sensitivity and the PV_−ve_ for CXCR4 was similar to those of CEA and higher than those of CXCR2 and CRP, while the AUC for CXCR4 was the highest among all the tested proteins (CXCR2, CEA, CRP). In reference to percentage of elevated concentrations, the combined analysis of CXCR4 and the well-established tumor marker may be more useful in the diagnosis of PC when compared to the measurement of a single biomarker. 

Our present study also revealed that serum CXCR2, CEA and age were significant predictors of PC risk. We have previously demonstrated that serum CXCL8 was the only significant predictor of PC risk [38].

## 5. Conclusions

The mortality rate of PC is equal to its incidence and the majority of PC patients die within a few months of diagnosis. According to our knowledge, this paper is the first study concerning the assessment of serum CXCR4 and CXCR2 in PC patients in relation to the well-established marker for this malignancy. Based on the presented findings, we may conclude that serum CXCR4 is a better candidate for a tumor marker than CXCR2 in the diagnosis of PC, while serum CXCR2 is a significant predictor of PC risk. 

## Figures and Tables

**Figure 1 ijms-21-06193-f001:**
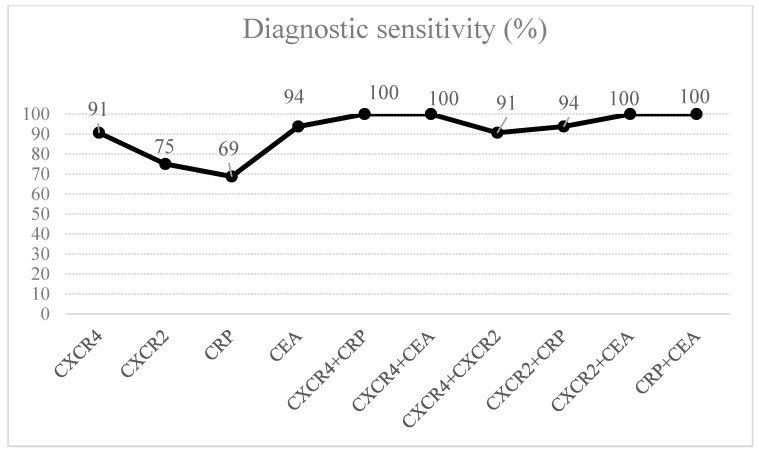
The percentage of elevated concentrations of CXCR4 and CXCR-2, well-established tumor marker (CEA) and C-reactive protein (CRP) in pancreatic cancer (PC) patients.

**Figure 2 ijms-21-06193-f002:**
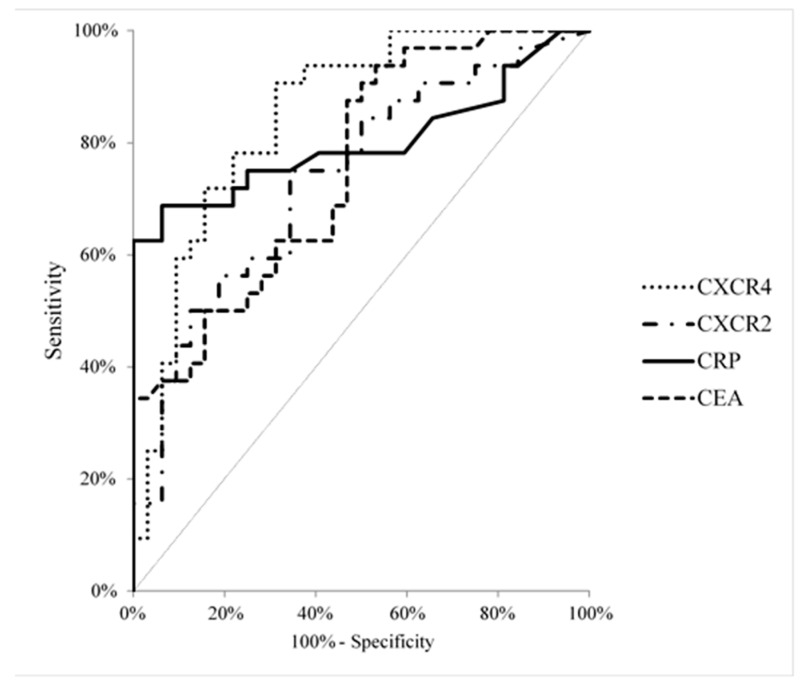
Areas under ROC curves (AUC) for CXCR4 (0.8496, *p* < 0.001), receptor CXCR2 (0.7373, *p* = 0.002), CEA (0.7539, *p* < 0.001), CRP (0.8013, *p* < 0.001) in pancreatic cancer patients.

**Table 1 ijms-21-06193-t001:** Characteristics of pancreatic cancer patients.

Variable Tested		Number of Patients
Group	Pancreatic cancer (PC)	32
Gender	Male	20
	Female	12
TNM Stage	I + II	8
	III	10
	IV	14
Depth of Tumor Invasion (T Factor)	T1 + 2 + 3	10
	T4	22
Nodal Involvement (N Factor)	N0	13
	N1	19
Distant Metastases (M factor)	M0	18
	M1	14

Abbreviations: M, distant metastases; N, nodal involvement; PC, pancreatic cancer; T, depth of tumor invasion; TNM, tumor nodulus-metastasis.

**Table 2 ijms-21-06193-t002:** Serum levels of analyzed proteins in pancreatic cancer patients in comparison to healthy controls.

Group Tested		CXCR-4 [ng/mL]	CXCR-2 [ng/mL]	CRP [mg/L]	CEA [ng/mL]
Pancreatic Cancer (PC)	Min	0.54	0.00	0.30	0.92
	Me	6.48	1.02	10.30	2.67
	Max	45.03	2.26	269.40	319.21
Control Group (Healthy Subjects)	Min	0.07	0.000	0.20	0.50
	Me	0.89	0.63	0.95	1.34
	Max	25.05	1.72	5.00	4.54
*p* (PC vs. Healthy Controls)		<0.001 ^a^	0.001 ^a^	<0.001 ^a^	<0.001 ^a^

^a^ Statistically significant when *p* < 0.05. Abbreviations: CEA, carcinoembryonic antigen; CRP, C-reactive protein; CXCR2, specific C-X-C motif chemokine receptor 2; CXCR4, specific C-X-C motif chemokine receptor 4; Max, maximum; Me, median; Min, minimum; PC, pancreatic cancer.

**Table 3 ijms-21-06193-t003:** Serum levels of analyzed proteins in relation to tumor stage of pancreatic cancer.

		CXCR4 [ng/mL]	CXCR2 [ng/mL]	CRP [mg/L]	CEA [ng/mL]
I + II	Min	1.70	0.00	0.30	1.20
	Me	3.95	1.03	12.95	1.72
	Max	45.03	1.86	161.50	10.94
III	Min	0.69	0.08	0.30	0.92
	Me	6.39	1.02	2.80	2.69
	Max	22.36	2.17	39.20	17.35
IV	Min	0.54	0.25	0.60	1.31
	Me	7.08	1.02	26.15	2.91
	Max	25.62	2.26	269.40	319.21
Control Group (CG)	Min	0.07	0.00	0.20	0.50
	Me	0.89	0.63	0.95	1.34
	Max	25.05	1.72	5.00	4.54
*p* (Kruskal-Wallis Test)		<0.001 ^a^	0.01 ^a^	<0.001 ^a^	0.02 ^a^
*p* (post hoc Dwass-Steele-Critchlow-Fligner test	I + II vs. III	0.98	1.00	0.63	0.76
	I + II vs. IV	0.99	1.00	0.76	0.29
	I + II vs. CG	0.01 ^a^	0.16	0.04 ^a^	0.49
	III vs. IV	0.90	1.00	0.10	0.90
	III vs. CG	0.01 ^a^	0.21	0.31	0.10
	IV vs. CG	<0.001 ^a^	0.03 ^a^	<0.001 ^a^	0.01 ^a^

^a^ Statistically significant when *p* < 0.05. Abbreviations: CEA, carcinoembryonic antigen; CRP, C-reactive protein; CXCR2, specific C-X-C motif chemokine receptor 2; CXCR4, specific C-X-C motif chemokine receptor 4; M, distant metastases; Max, maximum; Me, median; Min, minimum; N, nodal involvement; PC, pancreatic cancer; T, depth of tumor invasion; TNM, tumor-nodulus-metastasis.

**Table 4 ijms-21-06193-t004:** Serum levels of analyzed proteins in relation to clinicopathological features of PC.

Pancreatic Cancer (PC)			CXCR4 [ng/mL]	CXCR2 [ng/mL]	CRP [mg/L]	CEA [ng/mL]
Depth of tumor invasion (T factor)	T1 + 2 + 3	Min	1.70	0.00	0.70	1.20
		Me	5.88	0.93	12.65	1.99
		Max	45.03	1.86	161.50	10.94
	T4	Min	0.54	0.08	0.30	0.92
		Me	6.70	1.05	8.10	2.91
		Max	25.62	2.26	269.40	319.21
Control group (CG)		Min	0.07	0.00	0.20	0.50
		Me	0.89	0.63	0.95	1.34
		Max	25.05	1.72	5.00	4.54
*p* (Kruskal-Wallis test)			<0.001 ^a^	0.01 ^a^	<0.001 ^a^	0.01 ^a^
*p* (post hoc Dwass-Steele-Critchlow-Fligner test)		1 + 2 + 3 vs. 4	0.98	0.90	1.00	0.54
		1 + 2 + 3 vs.	0.01 ^a^	0.12	0.01 ^a^	0.10
		4 vs. CG	<0.001 ^a^	0.01 ^a^	0.01 ^a^	0.10
Presence of lymph node metastasis (N factor)	N0	Min	0.69	0.08	0.30	0.92
		Me	3.15	0.77	7.10	1.92
		Max	45.03	1.86	161.50	7.22
	N1	Min	0.54	0.00	0.30	1.24
		Me	6.74	1.06	14.30	3.00
		Max	25.62	2.26	269.40	319.21
Control group (CG)		Min	0.07	0.00	0.20	0.50
		Me	0.89	0.63	0.95	1.34
		Max	25.05	1.72	5.00	4.54
*p* (Kruskal-Wallis test)			<0.001 ^a^	0.01 ^a^	<0.001 ^a^	0.01 ^a^
*p* (post hoc Dwass-Steele-Critchlow-Fligner test)		0 vs. 1	0.47	0.64	0.36	0.25
		0 vs. CG	0.01 ^a^	0.11	0.01 ^a^	0.13
		1 vs. CG	<0.001 ^a^	0.01 ^a^	<0.001 ^a^	0.01 ^a^
Presence of distant metastasis (M factor)	M0	Min	0.69	0.00	0.30	0.92
		Me	5.78	1.03	6.75	1.99
		Max	45.03	2.17	161.50	17.35
	M1	Min	0.54	0.25	0.60	1.31
		Me	7.08	1.02	26.15	2.91
		Max	25.62	2.26	269.40	319.21
Control group (CG)		Min	0.07	0.00	0.20	0.50
		Me	0.89	0.63	0.95	1.34
		Max	25.05	1.72	5.00	4.54
*p* (Kruskal-Wallis test)			<0.001 ^a^	0.01 ^a^	<0.001 ^a^	0.01 ^a^
*p* (post hoc Dwass-Steele-Critchlow-Fligner test)		0 vs. 1	0.82	1.00	0.10	0.32
		0 vs. CG	<0.001 ^a^	0.03 ^a^	0.01 ^a^	0.04 ^a^
		1 vs. CG	<0.001 ^a^	0.02 ^a^	<0.001 ^a^	0.01 ^a^

^a^ Statistically significant when *p* < 0.05. Abbreviations: CEA, carcinoembryonic antigen; CRP, C-reactive protein; CXCR2, specific C-X-C motif chemokine receptor 2; CXCR4specific C-X-C motif chemokine receptor 4; M, distant metastases; Max, maximum; Me, median; Min, minimum; N, nodal involvement; PC, pancreatic cancer; T, depth of tumor invasion; TNM, tumor-nodulus-metastasis.

**Table 5 ijms-21-06193-t005:** Correlations between clinicopathological characteristics of pancreatic cancer and serum levels of analyzed proteins.

		T	N	TNM	Age	CXCR4	CXCR2	CRP	CEA
**T**	r	1.00	0.42	0.49	0.01	−0.04	0.06	0.02	0.22
	*p*		0.02 ^a^	<0.001 ^a^	0.97	0.84	0.74	0.90	0.24
**N**	r	0.42	1.00	0.53	0.00	0.21	0.16	0.24	0.29
	*p*	0.02 ^a^		<0.001 ^a^	0.99	0.25	0.38	0.18	0.11
**TNM**	r	0.49	0.53	1.00	0.33	0.04	−0.02	0.26	0.30
	*p*	<0.001 ^a^	<0.001 ^a^		0.06	0.81	0.90	0.15	0.09
**Age**	r	0.01	0.00	0.33	1.00	0.41	0.44	0.57	0.31
	*p*	0.97	0.99	0.06		<0.001 ^a^	<0.001 ^a^	<0.00 ^a^	<0.001 ^a^
**CXCR4**	R	−0.04	0.21	0.04	0.41	1.00	0.71	0.37	0.11
	*p*	0.84	0.25	0.81	<0.001 ^a^		<0.001 ^a^	<0.00 ^a^	0.39
**CXCR2**	R	0.06	0.16	−0.02	0.44	0.71	1.00	0.27	0.00
	*p*	0.74	0.38	0.90	<0.001 ^a^	<0.00 ^a^		0.03 ^a^	0.98
**CRP**	R	0.02	0.24	0.26	0.57	0.37	0.27	1.00	0.44
	*p*	0.90	0.18	0.15	<0.001 ^a^	<0.001 ^a^	0.03 ^a^		<0.001 ^a^
**CEA**	R	0.22	0.29	0.30	0.31	0.11	0.00	0.44	1.00
	*p*	0.24	0.11	0.09	0.01 ^a^	0.39	0.98	<0.001 ^a^	

^a^ Statistically significant when *p* < 0.05. Abbreviations: CEA, carcinoembryonic antigen; CRP, C-reactive protein; CXCR4, specific C-X-C motif chemokine receptor 4; CXCR2, specific C-X-C motif chemokine receptor 2; M, distant metastases; N, nodal involvement; T, depth of tumor invasion; TNM, tumor-nodulus-metastasis.

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
