# Peer review of "Specific Receptors for the Chemokines CXCR2 and CXCR4 in Pancreatic Cancer"

_ijms, 2020, doi:10.3390/ijms21176193_

Round 1
Reviewer 1 Report
I am satisfied with the revision and recommend it for publication.
Reviewer 2 Report
The authors have addressed all my comments/suggestions.
This manuscript is a resubmission of an earlier submission. The following is a list of the peer review reports and author responses from that submission.
Round 1
Reviewer 1 Report
The manuscript entitled “Specific receptors for chemokines-CXCR2 and CXCR4 in pancreatic cancer” evaluate the clinical significance of CXCR2 and CXCR4 levels in the serum of patients with pancreatic cancer (PC) for the diagnosis and the prediction of this deadly disease. Specifically, the authors compared the concentration of CXCR2 and CXCR4 with the classical markers CEA and CRP by using immunological methods and they found a significant increase of these markers in patients with PC respect to healthy controls. Moreover, the authors identified CXCR2 as a predictor of PC risk, while serum CXCR4 as a good candidate for a tumor marker.
The manuscript is well written, but I think that some minor revisions are required.
- As the authors known, there are various types of pancreatic cancer. The pancreatic ductal adenocarcinoma, which is the most common type of PC, and the neuroendocrine tumours, characterized by different symptoms, and they respond to different treatments with diverse outcome. Which kind of pancreatic cancer affected the patients enrolled in this study?
- The authors did not specify if the patients recruited have been experienced the chemotherapy treatment. If not, it would be interesting to analyze the association of CXCR2 and CXCR4 with response to chemotherapy.
- Could be possible to show the Kaplan-Meier curve performing the correlation between CXCR2 and CXCR4 levels with the overall survival?
Author Response
Department of Neurodegeneration Diagnostics, Medical University
Waszyngtona 15 a, 15-269 Białystok, Poland
phone 48 85 7468587, fax 48 85 7468585, e-mail: [email protected]
Białystok, 4th August 2020
Editor-in-Chief
International Journal of Molecular Science
Prof. Dr. Kurt A. Jellinger
Dear Editor,
Thank you very much for having our revised manuscript entitled: “Specific receptors for chemokines – CXCR2 and CXCR4 in pancreatic cancer”. The changes in the revised version of our manuscript have been highlighted (printed in bold). We hope you will find our answers satisfying and the revised manuscript will be acceptable for the publication.
In case of further questions, do not hesitate to contact me.
Sincerely,
Ala Litman-Zawadzka
DETAILED ANSWERS TO THE REVIEWERS’ COMMENTS
Reviewer 1
The manuscript entitled “Specific receptors for chemokines-CXCR2 and CXCR4 in pancreatic cancer” evaluate the clinical significance of CXCR2 and CXCR4 levels in the serum of patients with pancreatic cancer (PC) for the diagnosis and the prediction of this deadly disease. Specifically, the authors compared the concentration of CXCR2 and CXCR4 with the classical markers CEA and CRP by using immunological methods and they found a significant increase of these markers in patients with PC respect to healthy controls. Moreover, the authors identified CXCR2 as a predictor of PC risk, while serum CXCR4 as a good candidate for a tumor marker.
1. As the authors known, there are various types of pancreatic cancer. The pancreatic ductal adenocarcinoma, which is the most common type of PC, and the neuroendocrine tumours, characterized by different symptoms, and they respond to different treatments with diverse outcome. Which kind of pancreatic cancer affected the patients enrolled in this study?
Authors' Responses to Reviewer's Comments (Reviewer 1)
Thank you very much for positive review.
1. In the presented study patients with pancreatic ductal adenocarcinoma have been enrolled. The information concerning histological type of pancreatic cancer has been added in the Patients and Methods section, as it was recommended (page 2, line 78).
2. The authors did not specify if the patients recruited have been experienced the chemotherapy treatment. If not, it would be interesting to analyze the association of CXCR2 and CXCR4 with response to chemotherapy.
Authors' Responses to Reviewer's Comments (Reviewer 1)
Blood samples from PC patients were obtained before any treatment. Therefore, the sentence: ‘Our inclusion criteria were as follows: patients with PC (20 men and 12 women, aged 44-82 years) diagnosed in the Second Department of General Surgery, Medical University of Bialystok, Poland’ has been modified into: ‘Our inclusion criteria were as follows: patients with pancreatic ductal adenocarcinoma (20 men and 12 women, aged 44-82 years) before treatment diagnosed in the Second Department of General Surgery, Medical University of Bialystok, Poland’ (page 2, lines 78-79). However, it would be interesting to analyze the association of CXCR2 and CXCR4 levels with response to chemotherapy and these results might be presented in our forthcoming study, according to the Reviewer’s suggestion.
3. Could be possible to show the Kaplan-Meier curve performing the correlation between CXCR2 and CXCR4 levels with the overall survival?
Authors' Responses to Reviewer's Comments (Reviewer 1)
Thank you very much for the interesting suggestion. Our study was performed on patients with pancreatic ductal adenocarcinoma before treatment. Therefore, in presented paper, we are unable to add Kaplan-Meier curve in the Results section. However, we believe that Kaplan-Meier curve performing the correlation between CXCR2 and CXCR4 levels with the overall survival will be presented in our forthcoming study, as it was recommended.
Reviewer 2 Report
The study by Litman-Zawadzka et al. investigates the expression of CXCR2 and CXCR4 chemokine receptors in the serum of healthy patients or patients with pancreatic cancer and correlates this parameter with other clinical features of the patients. Overall, the measuring a transmembrane receptor in the serum appears atypical and requires an extensive justification and demonstration of the presence and location of such proteins. The state of the art on this field is not appropriately considered and discussed. In addition to this, I have several points to mention.
Major points
- CXCR2 and CXCR4 are transmembrane receptors. So, I don't really understand the rationale of measuring such proteins by Elisa in serum samples. Do the authors mean that these receptors are soluble and circulating proteins? This point is absolutely critical for the study and needs a clear demonstration of circulating CXCR2 and CXCR4 by other methods. Are the elisa kits used really specific?
- Line 38/39: CXCR2 is believed to be expressed by neutrophils and endothelial cells, not so much by monocytes. Moreover, CXCL8 is not the only ligand of CXCR2. CXCL1, 2, 3, 5, 6 and 7 are also very powerful activators of this receptor. This should be mentioned and cite paper from Strieter's group.
- The references cited line 40 are not adequate for some of them. The authors should cite studies on CXCR2 interaction with its ligands (for instance, papers from Rajarathnam group) and papers which have shown that CXCL8 and more generally CXCR2 ligands are overexpressed in aggressive breast cancers.
- lines 49 to 53: I don't understand the last sentence of this part, as the previous sentence deals with CXCR4 expression in fibroblasts
- line 58: there are a number of reviews on CXCR4 and PC that could be cited: for instance Slieghtholm et al. 2017, Pharmacol Ther
- the cohort of patients should be better described. Does it correspond to untreated patients?
- line 102: A better explanation of the cut-off used and the rationale of this should be given.
- line 125: the median of CXCR2 and CEA levels in PC does not seem very different from the one of healthy patients and still, the statistics seem as significant for CXCR2 and CEA as for CXCR34 and CRP. Can the authors explain this?
- line 137: the authors should mention that there were no differences of CXCR2 and CXCR4 between cancer patients whatever stage of the disease. I also don't understand why a distinct statistical was used in table 3 compared to table 2. What is the rationale? Why using the Dwass-Steele-Critchlow-Fligner test?
- line 147: the authors refer to tumor size in table 4, but the table mentions the depth of tumor invasion. Are these parameters exactly the same?
- in table IV the comparison 1+2+3 for T factor is vs ??? In the same table, I don't really understand the rationale of comparing the size of the tumor, presence of lymph node metastasis or of distant metastasis of cancer groups with control group. Only the comparison between cancer groups is relevant.
- figure 1: has the "diagnostic sensitivity" been performed also on healthy group as a negative control? This should be shown.
Minor points
- line 12: please define PC
- line 37: please cite the papers of Muller et al., 2001, Nature; Lazennec and Richmond, 2010, Trends Mol Med; Cheng 2019, BBA Rev Cancer
Author Response
Department of Neurodegeneration Diagnostics, Medical University
Waszyngtona 15 a, 15-269 Białystok, Poland
phone 48 85 7468587, fax 48 85 7468585, e-mail: [email protected]
Białystok, 4th August 2020
Editor-in-Chief
International Journal of Molecular Science
Prof. Dr. Kurt A. Jellinger
Dear Editor,
Thank you very much for having our revised manuscript entitled: “Specific receptors for chemokines – CXCR2 and CXCR4 in pancreatic cancer”. The changes in the revised version of our manuscript have been highlighted (printed in bold). We hope you will find our answers satisfying and the revised manuscript will be acceptable for the publication.
In case of further questions, do not hesitate to contact me.
Sincerely,
Ala Litman-Zawadzka
DETAILED ANSWERS TO THE REVIEWERS’ COMMENTS
Reviewer 2
Review Report Form
The study by Litman-Zawadzka et al. investigates the expression of CXCR2 and CXCR4 chemokine receptors in the serum of healthy patients or patients with pancreatic cancer and correlates this parameter with other clinical features of the patients. Overall, the measuring a transmembrane receptor in the serum appears atypical and requires an extensive justification and demonstration of the presence and location of such proteins. The state of the art on this field is not appropriately considered and discussed. In addition to this, I have several points to mention.
- Major points
- CXCR2 and CXCR4 are transmembrane receptors. So, I don't really understand the rationale of measuring such proteins by Elisa in serum samples. Do the authors mean that these receptors are soluble and circulating proteins? This point is absolutely critical for the study and needs a clear demonstration of circulating CXCR2 and CXCR4 by other methods. Are the elisa kits used really specific?
Authors' Responses to Reviewer's Comments (Reviewer 2)
Circulating CXCR2 and CXCR4 levels in patients were measured using enzyme-linked immunosorbent assay kits (ELISA) (Human C-X-C chemokine receptor type 4, CXC-R4 ELISA kit, EIAab, Cat. No: E2170h; Human C-X-C chemokine receptor type 2, CXC-R2 ELISA kit, EIAab, Cat. No: E2006h) according to the manufacturer’s instructions. The findings of other authors indicate that soluble human receptors for chemokines were indeed detected in the circulation of cancer patients. It is suggested that there are two in vivo mechanisms for the generation of soluble receptors are known: shedding of membrane-bound receptors and gene expression (Kalinkovich A, Bentwich Z. Soluble chemokine CCR5 receptor is present in human plasma. Immunology Letters. 2005; 96: 55-61). Moreover, presented paper is continuation of our previous studies, where we assessed the serum concentrations of specific receptors in patients with gastrointestinal malignancies (Łukaszewicz-Zając M, et al. Serum concentrations of receptor for interleukin 8 in patients with esophageal cancer. Pol Arch Intern Med. 2016;126:854-861; Łukaszewicz-Zając M, et al. The Serum Concentrations of Chemokine CXCL12 and Its Specific Receptor CXCR4 in Patients with Esophageal Cancer. Dis Markers. 2016; 7963895).
- Line 38/39: CXCR2 is believed to be expressed by neutrophils and endothelial cells, not so much by monocytes. Moreover, CXCL8 is not the only ligand of CXCR2. CXCL1, 2, 3, 5, 6 and 7 are also very powerful activators of this receptor. This should be mentioned and cite paper from Strieter's group.
Authors' Responses to Reviewer's Comments (Reviewer 2)
The sentence: “CXCR2 is present on monocytes and neutrophils” has been changed into: “CXCR2 is widely expressed by neutrophils and endothelial cells” (page 1, line 39). Moreover, the sentence: “The most potent ligand of this specific receptor is C-X-C motif chemokine 8 (CXCL8) and cleavage products of this chemokine [9-17].” has been modified into: “All ELR+ CXC chemokine ligands (CXCL1, CXCL2, CXCL3, CXCL5, CXCL6, CXCL7, CXCL8) bind to CXCR2. However, the most potent ligand of this specific receptor is C-X-C motif chemokine 8 (CXCL8) and cleavage products of this chemokine [9-21]” (page 1, lines 39-40). In addition, new reference no 21 has been added in the revised manuscript (Strieter RM, et al), according to Reviewer’s suggestion (page 1, line 42; page 10, lines 297-298).
- The references cited line 40 are not adequate for some of them. The authors should cite studies on CXCR2 interaction with its ligands (for instance, papers from Rajarathnam group) and papers which have shown that CXCL8 and more generally CXCR2 ligands are overexpressed in aggressive breast cancers.
Authors' Responses to Reviewer's Comments (Reviewer 2)
The references no 16 (Van Damme, et al.) and no 17 (Mazur , et al.) have been replaced by the paper of K. Rajarathnam et al. (reference no 19) and original paper concerning the assessment of CXCR2 ligands expression in aggressive breast cancers (reference no 20) in the revised manuscript, according to the Reviewer suggestion (page 1, line 42; page 10, lines 293-296).
- lines 49 to 53: I don't understand the last sentence of this part, as the previous sentence deals with CXCR4 expression in fibroblasts
Authors' Responses to Reviewer's Comments (Reviewer 2)
The sentences concerning the CXCR4 expression in fibroblasts as well as references no 19-23 concerning this issue have been removed from the revised manuscript, as it was recommended (page 2, line 48).
- line 58: there are a number of reviews on CXCR4 and PC that could be cited: for instance Slieghtholm et al. 2017, Pharmacol Ther
Authors' Responses to Reviewer's Comments (Reviewer 2)
New publication (Sleightholm RL, Neilsen BK, Li J. Emerging roles of the CXCL12/CXCR4 axis in pancreatic cancer progression and therapy. Pharmacol Ther. 2017; 179: 158-170) concerning the assessment of CXCR4 in PC patients has been added as reference no 24, in the revised manuscript, according to the Reviewer’s suggestion (page 2, line 55; page 11, lines 304-305).
- the cohort of patients should be better described. Does it correspond to untreated patients?
Authors' Responses to Reviewer's Comments (Reviewer 2)
Characteristics of patients has been presented in Table 1 (page 3, line 101). In addition, the information that PC patients before the treatment were included in our study has been added in Patients and Methods section, as it was recommended (page 2, line 79).
- line 102: A better explanation of the cut-off used and the rationale of this should be given.
Authors' Responses to Reviewer's Comments (Reviewer 2)
The most common criteria for cut-off values determination are the point on ROC curve where the sensitivity and specificity of the test are equal; the point on the curve with minimum distance from the left-upper corner of the unit square; and the point where the Youden’s index is maximum (Habibzadeh F, et al. On determining the most appropriate test cut-off value: the case of tests with continuous results. Biochem Med. 2016 Oct 15; 26(3): 297–307). However, the Perkins and Schisterman recommend the use of the Youden’s index and revealed that is the only “optimal” cut point for given weighting with respect to overall misclassification rates (Perkins NJ, Schisterman EF. The inconsistency of "optimal" cutpoints obtained using two criteria based on the receiver operating characteristic curve. Am J Epidemiol. 2006 Apr 1; 163(7):670-5). Therefore, the sentence: ‘In order to select optimal predicted probability cut-off values, the Youden's index was used’ has been modified into: ‘In order to select optimal predicted probability cut-off values, the Youden's index was used, because this index reflects the intension to maximize the correct classification rate’, as it was recommended (page 3, lines 98-99). In addition, the sentence: ‘The cut-off values of the analysed proteins were established using the Youden Index’ has been modified into: ‘The cut-off values of the analysed proteins were established using the Youden Index and were as follows: 0.72 ng/ml for CXCR2; 1.56 ng/ml for CXCR4; 3.20 mg/l for CRP and 1.24 ng/ml for CEA’ in the new version of paper, according to the Reviewer suggestion (page 7, lines 177-178).
- line 125: the median of CXCR2 and CEA levels in PC does not seem very different from the one of healthy patients and still, the statistics seem as significant for CXCR2 and CEA as for CXCR34 and CRP. Can the authors explain this?
Authors' Responses to Reviewer's Comments (Reviewer 2)
The data presented in Table 2 has been checked once again, and due to technical modifications of Table 2, the numbers have been shifted between rows and columns. Therefore, the data presented in Table 2 has been corrected in the revised version of manuscript , as it was recommended (page 3-4, line 126).
- line 137: the authors should mention that there were no differences of CXCR2 and CXCR4 between cancer patients whatever stage of the disease. I also don't understand why a distinct statistical was used in table 3 compared to table 2. What is the rationale? Why using the Dwass-Steele-Critchlow-Fligner test?
Authors' Responses to Reviewer's Comments (Reviewer 2)
In Table 2, the Mann-Whitney test was used to compare two groups (pancreatic cancer patients vs healthy subjects), while in Table 3, we used the Kruskal-Wallis test to analyze the associations between three or more groups (TNM I+II, TNM III, TNM IV and control group). In addition, the post hoc Dwass-Steele-Critchlow-Fligner test was performed if significant differences were calculated in Kruskal-Wallis test. Moreover, we need to present these results in two separate tables, because we believe that there would be too much data in one table. In addition, the sentence: ‘….we found that the serum levels of CXCR4, CRP and CEA were higher in advanced stage of disease in comparison to early PC. However, there were no differences of CXCR2 and CXCR4 levels between cancer patients and stage of the disease (Table 3).’ has been added in the new version of manuscript, according to Reviewer’s suggestion (page 4, lines 131-134).
- line 147: the authors refer to tumor size in table 4, but the table mentions the depth of tumor invasion. Are these parameters exactly the same?
Authors' Responses to Reviewer's Comments (Reviewer 2)
Abbreviation: ‘T factor’ has been unified as depth of tumor invasion in the revised version of manuscript, as it was recommended (page 4, line 142).
- in table IV the comparison 1+2+3 for T factor is vs ??? In the same table, I don't really understand the rationale of comparing the size of the tumor, presence of lymph node metastasis or of distant metastasis of cancer groups with control group. Only the comparison between cancer groups is relevant.
Authors' Responses to Reviewer's Comments (Reviewer 2)
Due to low number of patients in T1, T2 and T3 stage, we combined those subjects (T1+T2+T3) in one group (no=10) to perform the statistical analyses in accordance to T4 subgroup (no=22). In addition, the comparison of analyzed protein concentrations between depth of tumor invasion, presence of lymph node metastasis or of distant metastasis of cancer groups with control group has been omitted in the in the body of text of revised version of paper, according to Reviewer’s suggestion (page 4-5, lines 141-148).
- figure 1: has the "diagnostic sensitivity" been performed also on healthy group as a negative control? This should be shown.
Authors' Responses to Reviewer's Comments (Reviewer 2)
The percentage of elevated concentrations (diagnostic sensitivity) of analyzed proteins has been presented in the Figure 1. This diagnostic criteria has been calculated using the serum concentrations of analyzed proteins only in pancreatic cancer patients (study group), not in healthy group (negative control), according to formula: Sensitivity = a / a+c , where a (true positive) / a+c (true positive + false negative). In addition, this details have been presented in the title of Figure 1: ‘Figure 1. The percentage of elevated concentrations of CXCR4 and CXCR-2, well-established tumor marker (CEA) and C-reactive protein (CRP) in pancreatic cancer (PC) patients’, as it was recommended (page 7, lines 173-174).
- Minor points
- line 12: please define PC
Authors' Responses to Reviewer's Comments (Reviewer 2)
PC has been defined as pancreatic cancer, as it was recommended (page 1, line 12).
- line 37: please cite the papers of Muller et al., 2001, Nature; Lazennec and Richmond, 2010, Trends Mol Med; Cheng 2019, BBA Rev Cancer
Authors' Responses to Reviewer's Comments (Reviewer 2)
Three, new citations (Müller A, et; Lazennec G, et al; Cheng Y, et al.) have been added in the revised version of manuscript as references no 11-13, according to the Reviewer’s suggestions (page 1, line 37; page 10, lines 273-279).
Round 2
Reviewer 2 Report
The authors have made some improvements of the quality of the manuscript, but they don't really answer the key question of the presence of soluble chemokine receptors. Without a clear demonstration of their existence, it is difficult to assess the relevance of the findings.